# Predictive Markers for Immune Checkpoint Inhibitors in Non-Small Cell Lung Cancer

**DOI:** 10.3390/jcm11071855

**Published:** 2022-03-27

**Authors:** Ryota Ushio, Shuji Murakami, Haruhiro Saito

**Affiliations:** Kanagawa Cancer Center, Department of Thoracic Oncology, 2-3-2 Nakao, Asahi, Yokohama 241-8515, Japan; r-ushio@kcch.jp (R.U.); saito-h@kcch.jp (H.S.)

**Keywords:** non-small cell lung cancer, biomarker, anti-programmed cell death ligand 1, tumor-infiltrating lymphocytes, tumor mutation burden, human leukocyte antigen class I, DNA mismatch repair deficiency, microsatellite instability

## Abstract

Immune checkpoint inhibitors (ICIs) have dramatically improved the outcomes of non-small cell lung cancer patients and have increased the possibility of long-term survival. However, few patients benefit from ICIs, and no predictive biomarkers other than tumor programmed cell death ligand 1 (PD-L1) expression have been established. Hence, the identification of biomarkers is an urgent issue. This review outlines the current understanding of predictive markers for the efficacy of ICIs, including PD-L1, tumor mutation burden, DNA mismatch repair deficiency, microsatellite instability, CD8^+^ tumor-infiltrating lymphocytes, human leukocyte antigen class I, tumor/specific genotype, and blood biomarkers such as peripheral T-cell phenotype, neutrophil-to-lymphocyte ratio, interferon-gamma, and interleukin-8. A tremendous number of biomarkers are in development, but individual biomarkers are insufficient. Tissue biomarkers have issues in reproducibility and accuracy because of intratumoral heterogeneity and biopsy invasiveness. Furthermore, blood biomarkers have difficulty in reflecting the tumor microenvironment and therefore tend to be less predictive for the efficacy of ICIs than tissue samples. In addition to individual biomarkers, the development of composite markers, including novel technologies such as machine learning and high-throughput analysis, may make it easier to comprehensively analyze multiple biomarkers.

## 1. Introduction

Lung cancer is the most frequent cause of cancer death worldwide. In 2020, 2.21 million new cases (11.4% of all cancer cases) and 1.80 million deaths (18.0% of all cancer deaths) were reported [1]. The most common histological type is non-small cell lung cancer (NSCLC), and most patients are diagnosed at an advanced stage [2]. Platinum-based chemotherapy has historically been the standard treatment for NSCLC, although limited therapeutic effects in patients with a poor prognosis have been observed. Recently, the advent of immune checkpoint inhibitors (ICIs) such as nivolumab and pembrolizumab (anti-programmed cell death 1 (PD-1) antibodies), atezolizumab and durvalumab (anti-programmed cell death ligand 1 (PD-L1) antibodies), and ipilimumab and tremelimumab (anti-cytotoxic T-lymphocyte-associated antigen 4 (CTLA-4) antibody) has dramatically altered the approach of advanced NSCLC treatment. First-line ICI therapy has demonstrated more prolonged survival than conventional platinum-based chemotherapy for stage IV NSCLC. In a phase III trial (KEYNOTE-024), pembrolizumab increased overall survival (OS) to 30 months for NSCLC patients with a PD-L1 tumor proportion score (TPS) > 50%, thereby demonstrating its superiority to conventional platinum-based chemotherapy [3]. Furthermore, a phase III trial (KEYNOTE-042) comparing pembrolizumab monotherapy with platinum-based combination therapy in NSCLC patients with PD-L1 TPS ≥ 1% showed significantly longer OS in the pembrolizumab group than in the chemotherapy group [4]. Atezolizumab also prolonged OS over platinum-based chemotherapy (17.5 vs. 15.1 months) in advanced NSCLC with PD-L1 ≥ 1% of tumor cells (TCs) or ≥ 1% of tumor-infiltrating immune cells (ICs), regardless of histology [5]. In a phase III study (CheckMate 227), nivolumab plus ipilimumab was associated with better OS than chemotherapy (17.1 vs. 14.9 months) in patients with NSCLC, regardless of PD-L1 expression level [6]. First-line ICIs combined with chemotherapy are among the current standard therapies for advanced NSCLC, compensating for the disadvantages of early treatment failure with ICI monotherapy. In a phase III trial (KEYNOTE-189) of patients with advanced NSCLC, the addition of pembrolizumab to platinum-doublet chemotherapy significantly prolonged PFS and OS (survival rate at 12 months was 69.2% vs. 49.4%) [7]. In addition, this combination therapy overcame the early failure of ICIs, which had been a problem with single-agent therapy [7]. In a phase III trial (KEYNOTE-407) evaluating the efficacy of pembrolizumab added to platinum-based combination therapy in patients with advanced lung squamous cell carcinoma (LUSC), OS was significantly prolonged (17.1 vs. 11.6 months) [8]. A phase III study (IMpower150) revealed that the addition of atezolizumab to carboplatin/paclitaxel or carboplatin/paclitaxel/bevacizumab in non-squamous NSCLC patients significantly prolonged OS to 19.2 months [9]. Furthermore, in another phase III study (IMpower130), atezolizumab added to carboplatin/nab-paclitaxel significantly prolonged OS to 18.6 months [10], while in the CheckMate 9LA phase III trial, nivolumab/ipilimumab in combination with platinum-based therapy significantly prolonged OS to 15.6 months compared with platinum-based therapy in NSCLC patients [11]. ICIs also raised the possibility that advanced NSCLC patients may have a better chance of long-term survival. A first-line phase III immunotherapy trial (KEYNOTE-024) with pembrolizumab for NSCLC achieved a 5-year OS rate of 31.9% [12]. Two phase III trials of nivolumab (CheckMate 017 and CheckMate 057) in patients with previously treated advanced NSCLC demonstrated 5-year OS rates of 13.4% and 8.0%, respectively [13].

However, many NSCLC patients do not benefit from ICIs or suffer significant life-threatening immunotoxicity [14]. Immune-related adverse events (ir-AEs) can affect various organs, and dermatitis, pneumonitis, colitis, and endocrinopathies tend to be most common. While most cases are mild to moderate in severity, some cases are severe or even fatal, especially when not promptly recognized and appropriately managed [15,16]. Although it is essential to administer ICIs to appropriate patients, the expression of programmed death-ligand 1 (PD-L1), a widely used biomarker, is not a sufficient predictive factor. ICIs are effective even in NSCLC patients with low or absent PD-L1 expression and may not be effective in patients with high PD-L1 expression. Therefore, there is an urgent need to identify new biomarkers to predict the response to ICIs for selecting the best anti-cancer agents for each patient. This article reviews the biomarkers currently under development for ICIs in NSCLC.

## 2. Programmed Death-Ligand 1

As is known, PD-1 is a receptor on the surface of activated T and B cells that binds to PD-L1 and programmed death-ligand 2 (PD-L2). Furthermore, PD-1 binding to PD-L1 inhibits the cytotoxic/cytolytic effector function of T cells. This process is also known as T-cell exhaustion [17]. Additionally, PD-L1 is also expressed in tumor cells, suppressing the host’s immune response, and leading to tumor tolerance.

Currently, PD-L1 TPS is the most widely used biomarker for the efficacy of ICIs, and its therapeutic effect is dependent on the intensity of its expression. In the KEYNOTE-010 study comparing pembrolizumab with docetaxel for advanced previously treated NSCLC patients, as measured by the Dako 22C3 immunohistochemistry (IHC) assay, the HR for death of patients with TPS ≥ 50% was 0.53, which was lower than the HR of 0.76 for TPS 1–49% [18]. In the KEYNOTE-189 study evaluating the efficacy of adding pembrolizumab to chemotherapy using the Dako 22C3 IHC assay, pembrolizumab also demonstrated therapeutic effects according to the level of PD-L1 expression, and, the 12-month OS was 61.7% vs. 52.2% in patients with PD-L1 TPS < 1%, 71.5% vs. 50.9% for PD-L1 TPS 1–49%, and 73.0% vs. 48.1% in patients with PD-L1 TPS ≥ 50% [7]. Furthermore, PD-L1 TPS “high” expression is usually defined as ≥ 50%; however, patients with “very high” PD-L1 expression are expected to benefit to a greater extent from ICIs. A retrospective analysis comparing the effects of first-line pembrolizumab in NSCLC patients with PD-L1 TPS ≥ 90% and with 50–89% expression demonstrated significantly prolonged ORR (60.0% vs. 32.7%), PFS (14.5 vs. 4.1 months), and OS (not reached vs. 15.9 months, HR 0.39) in patients with PD-L1 TPS ≥ 90% [19]. In the OAK study comparing atezolizumab with docetaxel in previously treated NSCLC patients, as measured by Ventana SP142, the subgroup with PD-L1 expression on ≥50% of TCs or ICs benefited most from atezolizumab; OS and HR were 20.5 months and 0.41, respectively, and 8.9 months and 0.75 for patients without PD-L1 expression on TCs or ICs, respectively [20].

Four PD-L1 IHC assays are currently registered with the U.S. Food and Drug Administration (FDA), each of which uses a different PD-L1 antibody (22C3, 28-8, SP263, and SP142) and two different IHC platforms (Dako and Ventana). Unlike the 22C3, 28-8, and SP263 antibodies, which assess PD-L1 expression only on tumor cell membranes, the SP142 antibody assay used for atezolizumab measures PD-L1 levels in both TCs and ICs. Three reports evaluating four PD-L1 IHC assays in NSCLC tumor cells suggest good concordance between the Ventana SP263, Dako 28-8, and Dako 22C3 assays, whereas SP142 was found to be less sensitive [21,22,23].

In addition to the variability of PD-L1 antibodies and platforms, the PD-L1 analysis has some limitations. The evaluation of formalin-fixed tissue compared with fresh-frozen tissue may underestimate PD-L1 expression [24]. Furthermore, a small tissue sample may not be representative of the characteristics of the entire tumor because of tumor heterogeneity. Ilie et al. analyzed 160 NSCLC patients and reported a poor association of PD-L1 expression in TCs and ICs between lung biopsies and corresponding resected tumors (overall discordance rate = 48) using the SP142 IHC assay [25]. Kim et al. also reported that seven out of fifty NSCLC patients (14%) demonstrated discordance in PD-L1 expression tissue microarray specimens and the corresponding resected specimens using the 22C3 IHC assay [26]. Furthermore, PD-L1 expression may vary depending on the organ from which it is obtained. Uruga et al. analyzed 109 LUAD patients and reported a 38% discordance in PD-L1 expression between the primary tumor and nodal metastases [27]

## 3. Predictive Biomarkers beyond PD-L1 Expression

### 3.1. Tumor Mutational Burden

Tumor mutational burden (TMB) is defined by the number of mutation calls (somatic single variant (SNV) and multinucleotide variant (MNV) and small insertions and deletions (indels)) per megabase (Mb) of interrogated coding sequences. These mutations can be transcribed and translated into neoantigen-containing peptides, processed by the antigen-processing machinery, and loaded onto major histocompatibility complex (MHC) molecules for presentation on the cell surface. The immune system recognizes neoantigens as non-self-immunogenic targets, activating and targeting T cells [28,29,30,31]. Tissue and blood TMB is a potential biomarker of immunotherapy outcomes in multiple tumor types. In particular, lung cancer is primarily caused by chronic exposure to carcinogens in cigarette smoke, and the efficacy of ICIs correlates with a molecular signature characteristic of cigarette carcinogen-related mutagenesis, certain DNA repair mutations, and the burden of neoantigens [32,33]. An analysis of the CheckMate 568 study of nivolumab plus ipilimumab in NSCLC reported that ORR increased in patients with a higher tissue tumor mutational burden (tTMB) using the FoundationOne CDx (F1CDx) assay, plateaued at a threshold of 10 mutations (mut)/Mb (ORR: 4%, 10%, 44%, and 39% in patients with TMB <5, <10, 10, and ≥15 mut/Mb, respectively), and the enhanced response was independent of PD-L1 expression [34]. The CheckMate 227 study of nivolumab plus ipilimumab in NSCLC also demonstrated longer PFS in patients with tTMB-high, with at least 10 mut/Mb, irrespective of tumor PD-L1 expression level [35], while a phase III trial of durvalumab (MYSTIC) and tremelimumab indicated longer PFS and OS in NSCLC patients with blood TMB (bTMB)-high, with at least 20 mut/Mb [32]. A retrospective analysis of the POPLAR and OAK studies demonstrated that a positive correlation of tTMB and bTMB and NSCLC patients with bTMB ≥ 16 mut/Mb led to an increased PFS benefit from atezolizumab [36].

Although TMB is optimally calculated by whole-exome sequencing (WES), this approach presents difficulties in terms of its substantial cost and turnaround time in clinical settings [37]. To address this, targeted sequencing assays enriched with known cancer-driving gene mutations are used to assess TMB. The F1CDx assay and Memorial Sloan Kettering-Integrated Mutation Profiling of Actionable Cancer Targets (MSK-IMPACT) assay are moderately concordant with WES in TMB analysis [38,39], and both assays were recently approved as companion diagnostics by the FDA to assess TMB in solid tumors. In this approval, tTMB-high was defined as having at least 10 mut/Mb according to the KEYNOTE-158 study of pembrolizumab for unresectable or metastatic solid tumors [40]. Despite these initial positive findings, the role of TMB as a biomarker in NSCLC remains unclear. It was reported that tTMB was not predictive of the efficacy of pembrolizumab alone or in combination with chemotherapy according to retrospective analyses of the KEYNOTE-189 and KEYNOTE-021 studies, respectively [41,42].

TMB as a biomarker has other limitations, including a lack of standardization between the testing platforms. Low tumor purity may lead to inaccurate TMB estimates [43]. Lung cancer specimens often have a low tumor cell content due to inflammatory cells and stromal components, leading to an underestimation of TMB. Furthermore, although high-TMB is thought to lead to increased neoantigens, the effect on the tumor immune response may vary depending on whether the neoantigen is derived from clonal (or homogeneous tumor) or subclonal (or heterogeneous tumor) mutations because the lower antigen dosage compared with the clonal neoantigen burden reduces the chances of identifying T cells reactive to subclonal neoantigens. McGranahan et al. demonstrated that sensitivity to ICIs in NSCLC and melanoma patients was enhanced in tumors enriched for clonal neoantigens, and cytotoxic chemotherapy-induced subclonal neoantigens were enriched in certain poor responders [44].

### 3.2. DNA Mismatch Repair Deficiency and Microsatellite Instability

DNA mismatch repair (MMR) is a highly conserved biological DNA repair pathway in mammalian cells and is crucial for maintaining genomic stability. MMR deficiency (dMMR) is the initiating event in many cancer types [45]. The deficient DNA MMR mechanism leads to missed DNA replication errors, resulting in the increased acquisition of mutations, primarily in the form of microsatellite instability (MSI) or alterations in microsatellites, which increases the burden of neoantigens [46,47].

A clinical trial of pembrolizumab across dMMR tumors spanning 12 cancer types demonstrated that ORR was 53%, and complete response (CR) was 21% [48]. This led to pan-cancer approval by the FDA. However, the role of the MMR status as a predictive biomarker for immunotherapy in lung cancer remains unknown because it was not included in this study. The prevalence of MSI-high (MSI-H) status is rare, at 0.53% and 0.60% of lung adenocarcinomas (LUAD) and LUSCs, respectively [49].

### 3.3. CD8^+^ Tumor-Infiltrating Lymphocytes

The adaptive immune system identifies and targets tumor cells. Interestingly, CD8^+^ T cells, CD4^+^ T cells, B cells, dendritic cells, and effectors of innate immunity, namely macrophages, polymorphonuclear leukocytes, and natural killer cells (NK), as well as all cell types within the tumor, are classified as tumor-infiltrating lymphocytes (TILs) [50,51]. Of them all, the presence of tumor-infiltrating CD8^+^ T cells, which recognize tumor antigens, is a prerequisite for successful ICI treatment when presented at the tumor cell surface in the context of HLA class I. Several small-sized studies and a meta-analysis demonstrated that CD8^+^ TILs were significantly associated with better OS, PFS, and ORR in NSCLC patients treated with ICIs [52,53,54,55,56,57,58]. Shirasawa et al. demonstrated a classification system based on PD-L1 expression and CD8^+^ TIL status that accurately predicts the efficacy of ICIs in NSCLC patients better than tumor PD-L1 expression [59]. Furthermore, Kumagami et al. showed that the frequency of PD-1^+^CD8^+^ T cells relative to that of PD-1^+^ regulatory T (Treg) cells in the tumor microenvironment could predict the efficacy of ICIs more accurately than tumor PD-L1 expression [60].

Although TILs have great predictive power, they present some technical problems as biomarkers in clinical practice, and CD8^+^ TILs within the stroma and invasive margin compartment indicate a better outcome than those in the intratumoral compartment [56,61]. However, biopsy samples obtained by CT-guided needle biopsy or bronchoscopy in patients with advanced NSCLC are often insufficient to evaluate stromal TILs, and a sample may not represent the TME of the entire primary tumor or metastatic lesions.

### 3.4. Human Leukocyte Antigen Class I

The human leukocyte antigen (HLA) system encodes cell-surface proteins involved in immune system regulation [62]. Furthermore, HLA-I presents peptides derived from intracellular proteins on the surface of CD8^+^ T cells, so that cancer cells are killed [63,64].

Several specific HLA-I genotypes are suggested as biomarkers for ICI treatment; Naranbhai et al. reported that HLA-A*03 alleles led to a low ORR and poor PFS and OS in various cancer patients, including NSCLC with ICI treatment, in the most significant epidemiological analysis of the association between HLA-I and ICI efficacy so far [65]. In melanoma patients, HLA-B44 supertype and HLA-A02 supertype led to prolonged OS with ICI treatment [66,67]. Losses in heterozygosity (LOH) and *HLA* gene expression have also been reported as candidate biomarkers. Chowell et al. reported that LOH in cancer reduces OS in NSCLC and melanoma patients with ICI treatment [66,67]. However, Negrao et al. reported no significant correlations between HLA-I zygosity and PFS or OS in NSCLC patients with ICI treatment [68]. Schaafsma et al. analyzed 33 cancer types and reported that tumors with high *HLA* gene expression tended to have higher immune cell infiltration, including CD8^+^ T and NK cells, and a more immunologically active TME, thus leading to increased survival [69].

### 3.5. Blood Biomarkers

#### 3.5.1. Peripheral T-Cell Phenotype

Surface and intracellular proteins expressed on T cells are expected to be biomarkers because ICIs target T-cell regulatory pathways. A prominent surface marker mainly expressed by CD8 effector memory T-cells is PD-1 [70]. Interestingly, PD-1 on CD8 TILs is used as a marker of tumor-reactive cells [71]. Indeed, peripheral blood PD-1^+^CD8 T-cells can also express neo-antigen-recognizing T-cell receptors [72].

An analysis of 29 NSCLC patients treated with PD-1 inhibitor demonstrated that 70% of patients with disease progression lacked a PD-1^+^CD8 T-cell response, whereas 80% of patients with a clinical response showed PD-1^+^CD8 T-cell responses within 4 weeks from the induction of treatment Kamphorst et al. [73].

Furthermore, CX3C chemokine receptor 1 (CX3CR1) is a receptor of the chemokine CX3CL1, which is involved in the adhesion and migration of leukocytes [74,75]. Furthermore, CX3CR1 is a marker of T-cell differentiation and is rigidly expressed on CD8^+^ T cells through irreversible differentiation from CX3CR1^−^CD8^+^ T cells during the effector phase [75,76], which theoretically provides an advantage as a biomarker compared with transiently expressed molecules on T cells. Yamaguchi et al. reported that an increase in the frequency of the CX3CR1^+^ subset in circulating CD8^+^ T cells early after ICI therapy correlated with response and survival in 36 NSCLC patients [77].

The CD62L^low^ T-cell subpopulation in tumor-draining lymph nodes contains antitumor T cells and mediates potent antitumor activity when intravenously transferred [78,79]. Kgamu et al. reported that patients who responded to ICIs had a significantly higher ratio of effector CD62L^low^ CD4^+^ T cells in their peripheral blood before treatment, and that a decreased CD62L^low^ CD4^+^ T-cell ratio after ICI treatment resulted in resistance, with long-term survivors maintaining a high proportion of CD62L^low^ CD4^+^ T-cells [80].

#### 3.5.2. Neutrophil-to-Lymphocyte Ratio

The neutrophil-to-lymphocyte ratio (NLR) is a surrogate marker of general host immune response to various stress stimuli. The systemic inflammatory response of cancer prompts neutrophil infiltration, resulting in the secretion of interleukin-2 (IL-2), interleukin-6 (IL-6), interleukin-10 (IL-10), tumor necrosis factor α (TNF-α), and vascular endothelial growth factor (VEGF) [81]. The activation and intratumoral invasion of lymphocytes are thought to be necessary for the antitumor activity of ICIs [82], while TNF-α and IL-10 cause lymphocyte dysfunction and a decrease in lymphocyte numbers [83,84].

A meta-analysis reported that a high NLR resulted in a worse OS and PFS in melanoma, NSCLC, and genitourinary cancer treated with ICIs [85]. Several retrospective studies and one prospective study reported that high NLR correlated with the poor prognosis of NSCLC patients [86,87]. Furthermore, a retrospective study analyzing 466 NSCLC patients treated with ICIs demonstrated that NLR > 3 combined with an LDH of over the upper limit of normal resulted in poor OS and PFS for patients with ICI treatment, but not for patients with chemotherapy [88]. Hence, Mezquita et al. developed the lung immune prognostic index (LIPI) calculated by pre-treatment LDH and dNLR for use in selecting ICI treatment in lung cancer patients [88].

#### 3.5.3. Interferon-Gamma

Interferon-gamma (IFN-γ) is a cytokine that plays a role in innate and adaptive immunity, and is produced predominantly by T cells and NK cells for innate immunity but also by CD4^+^ and CD8^+^ T cells for adaptive immunity [89]. In tumors, TILs are the primary source of IFN-γ [90]. IFN-γ engages JAK/STAT signaling in the tumor cell, which induces MHC class I expression, accumulates effector cells, and promotes a loss of the suppressive activity of T-regs. Moreover, the deficiency of INF-γ inhibits effective innate and adaptive antitumor immunity [91,92].

Some reports suggested that a high-INF-γ level is related to the efficacy of ICIs. A study of durvalumab for previously treated NSCLC patients demonstrated that patients with a high pre-treatment IFN-γ signature (high levels of *IFN-γ*, *LAG3*, *CXCL9*, and *PD-L1* mRNA expression) had higher ORR, PFS, and OS [93]. An analysis of 17 NSCLC patients treated with nivolumab showed a trend of a more prolonged OS in those with high INF-γ expression compared with those with low *INF-γ* expression [94]. In the POPLAR study evaluating atezolizumab in NSCLC patients, a high T-effector-IFNγ-associated gene expression status was correlated with prolonged OS [95].

#### 3.5.4. Interleukin-8

As is known, interleukin-8 (IL-8) is a member of the CXC chemokine family and was initially identified as a chemotactic factor for neutrophils [96]. Furthermore, IL-8 is secreted by malignant cells and tumor stroma cells across many different tumor types [97]. Moreover, IL-8 directly affects endothelial cells, malignant cells, and cancer stem cells, and indirectly affects attracting and modulating tumor-associated myeloid cells [98,99].

Sanmamed et al. reported early increases in serum IL-8 levels as a predictor of poor outcome in small retrospective cohorts of patients with advanced melanoma or NSCLC who received ICIs [100]. A retrospective analysis demonstrated that high levels of IL-8 at the initiation of ICIs led to a poorer OS across renal cell carcinoma, melanoma, NSCLC, and urothelial cancer [101]. However, a poorer OS was also observed for other factors aside from ICI treatment, suggesting that IL-8 may also be a prognostic marker rather than a predictive biomarker of ICI treatment.

#### 3.5.5. Blood/Tissue Composite Biomarker

Nebet et al. demonstrated that pre-treatment circulating tumor DNA (ctDNA) and peripheral CD8 T cell levels are independently associated with the durable clinical benefit of ICIs, and developed the DIREct-Pre (durable immunotherapy response estimation by immune profiling and ctDNA pre-treatment) score system, combining tumor PD-L1 expression with pre-treatment ctDNA and circulating immune cell profiling [102]. Patients with higher DIREct-Pre scores had significantly longer PFS with ICIs [102].

## 4. Tumor/Specific Genotype

Molecular genetic analyses of lung cancer have recently become the standard of care for treatment selection, especially in LUAD. With the recent developments in genetic analysis technologies such as next-generation sequencing, molecular targeting therapy has been dramatically enhanced. The Lung Cancer Mutation Consortium reported that actionable drivers were detected in 64% of LUAD patients, and the OS of 260 patients with driver gene mutations and treatment with molecular targeted therapy was 3.5 years, while that of 318 patients with a driver and no molecular targeted therapy was 2.4 years [103]. Molecular-targeted agents are critical drugs and are generally more important than ICIs for NSCLC patients with driver gene mutations. However, the positioning and characteristics of ICIs vary with each gene.

### 4.1. EGFR Mutation

*EGFR* mutation is the most frequently detected oncogenic driver in Asian populations with LUAD, with 50% incidence and 10–15% incidence in western patients [104]. The efficacy of ICIs is generally poor for *EGFR*-mutated lung cancer patients, while EGFR-TKIs are effective. A meta-analysis of subgroups in phase III trials revealed a lower efficacy of ICIs for *EGFR*-mutated NSCLC than for *EGFR* wild-type NSCLC [105,106]. A small phase II trial that investigated the efficacy of pembrolizumab for *EGFR*-mutated NSCLC did not demonstrate a response or achieve a sustained duration of treatment for 1 year, even though it was restricted to PD-L1-positive patients, of whom 73% had a PD-L1 expression of ≥50% [107]. In a phase II trial (ATLANTIC) evaluating durvalumab in *EGFR*-/*ALK*-mutated NSCLC, the ORR was 3.6% in PD-L1 < 25% and 12.2% in PD-L1 ≥ 25% patients [108].

Although the reasons for these poor benefits have yet to be elucidated, Ganior et al. reported that *EGFR*-/*ALK*-mutated patients had low PD-L1 expression and a low presence of CD8^+^ TILs, which was described as a “cold” tumor microenvironment [109]. A low TMB may also explain the limited efficacy of ICIs in *EGFR*-/*ALK*-mutated lung cancer. *EGFR* mutations are common in light and never smokers, and The Cancer Genome Atlas revealed that the TMB in the *EGFR*-mutated group was comparatively lower than that in *EGFR* wild-type patients [110].

However, *EGFR* mutation-positive lung cancer cases are diverse, with some likely to benefit from ICI treatment. Haratani et al. reported subgroups of *EGFR*-mutated NSCLC patients with PD-L1 expression and high TMB, or that had acquired resistance to EGFR-TKIs by a mechanism other than the development of *T790M* mutation, possibly achieving a treatment benefit from ICIs [111]. Furthermore, some exploratory subgroup analyses suggested the efficacy of atezolizumab added to chemotherapy for *EFGR*-mutated NSCLC [10,112]. In a phase III trial (IMpower 150) comparing carboplatin/paclitaxel/bevacizumab/atezolizumab with carboplatin/paclitaxel/bevacizumab for *EGFR*-mutated NSCLC patients, median OS was not reached vs. 18.7 months OS (HR 0.61), and median PFS was 10.2 vs. 6.9 months, respectively [112]. In a phase III trial (IMpower 130) comparing carboplatin/nab-paclitaxel/atezolizumab with carboplatin/nab-paclitaxel for *EGFR*/*ALK*-mutated non-squamous non-small cell lung cancer, median PFS was 7.0 vs. 6.0 months, and median OS was 14.4 vs. 10.0 months, respectively [10].

Adverse events resulting from ICIs in *EGFR*-mutated lung cancer patients should be monitored. For example, an increased pulmonary toxicity in patients treated with osimertinib after receiving ICIs was reported [113,114], and phase I studies of durvalumab and osimertinib were stopped early because of an elevated incidence of pneumonitis [115]. Increased liver toxicity in patients treated with osimertinib after receiving ICIs was also reported [116]. In addition, pembrolizumab in combination with gefitinib therapy showed severe liver toxicity in five of seven patients in the KEYNOTE-021 study [117].

### 4.2. ALK Rearrangement

*ALK* rearrangement is an oncogenic driver that occurs in 5–6% of LUAD, and is prevalent in young non-smokers [118,119]. In contrast to the potent efficacy of ALK-TKIs, there are few reports on the effectiveness of ICIs for *ALK*-rearranged lung cancer. In a phase II trial (ATLANTIC), no responses were observed in 15 *ALK*-rearranged lung cancer patients treated with durvalumab, regardless of PD-L1 expression [109]. In the IMMUNOTARGET study, all 23 *ALK*-mutated patients did not respond to ICI treatment, and achieved PFS and OS rates of 2.5 months and 17 months, respectively [120]. A retrospective study reported no responses in the six *ALK*-rearranged lung cancer patients evaluated [109], while in a retrospective multicenter French study, only two of eight *ALK*-rearranged patients achieved a response [121].

Notably, a higher risk of hepatic toxicities has been reported with the sequential use of ALK-TKs after ICIs. In a retrospective study, five of eleven patients treated with crizotinib after ICIs developed severe liver dysfunction, corresponding to an incidence of 45.5% compared with 8% of those receiving crizotinib alone, which led to permanent discontinuation of crizotinib in four of five patients.

### 4.3. KRAS Mutation

*KRAS* is the most common oncogenic driver in NSCLC and is identified in up to 25% of adenocarcinomas and 3% of squamous cell carcinomas in western countries, and its frequency increases in smokers [122,123,124]. It has been reported that *KRAS* mutation induces PD-L1 expression via phosphorylated extracellular signal-regulated kinase (p-ERK) signaling in lung adenocarcinoma [125]. Among them, *KRAS p.G12C* patients were most likely to be PD-L1-positive, with 65.5% TPS ≥ 1% and 41.3% TPS ≥ 50% [126].

The impact of *KRAS* gene mutation on the efficacy of ICIs varies among reports. However, *KRAS* mutation is not a negative biomarker for ICI efficacy, while other oncogenic driver mutations are negative. A prospective study that evaluated ICIs for *KRAS*-mutated NSCLC patients demonstrated that *KRAS* mutation was an independent predictor of the long-term efficacy of ICIs [127]. Three retrospective studies suggested that *KRAS* mutation is predictive of a superior response to immunotherapy [127,128,129]. However, two retrospective studies comprising 162 and 59 *KRAS*-mutated NSCLC patients reported that ICI efficacy is independent of the existence of *KRAS* gene mutations [130,131].

### 4.4. BRAF Mutation

*BRAF* mutations are found in 2–4% of patients with non-squamous NSCLC, and *BRAF^V600E^* mutation accounts for 50% of these. [132]. Dudnik et al. reported that *BRAF*-mutated NSCLC tends to present high PD-L1 expression without high TMB or MSI-high, and PFS was 3.7 months and 4.1 months in *BRAF^V600E^*-mutated and non-*BRAF^V600E^*-mutated patients with ICI treatment, respectively [133]. In a retrospective study of 37 *BRAF*-mutated lung cancer patients, ORR was 24% and PFS was 3.1 months, which was slightly better than that of patients with *EGFR* mutation [121]. Moreover, the treatment of *BRAF*-mutated lung cancer with ICIs showed some efficacy, reflecting high PD-L1 expression and TMB.

### 4.5. MET Exon 14 Skipping

Interestingly, *MET* exon 14 skipping alterations occur in 3–4% of NSLCL, and MET inhibitors are active in patients with advanced *MET* exon 14-altered lung cancers [134]. Sabari et al. analyzed 147 lung cancer patients with *MET* exon 14 skipping; 41% of patients had more than 50% PD-L1 expression, and the median TMB was relatively low compared with overall NSCLC [135]. The ORR of ICI monotherapy was 17%, and the median PFS was 1.9 months. This result suggests that the efficacy of ICIs for *MET* exon 14-altered lung cancers is modest despite the frequency of high-PD-L1 expression [135].

### 4.6. STK11/LKB1 and KEAP1 Mutations

The prevalence of *STK11/LKB1* mutation is approximately 8–21% in LUAD and 1.5–5% in LSCC [136], and *STK11/LKB1* is co-mutated in 25.4% of *KRAS*-mutated LUAD [137]. *KEAP1* mutation prevalence in NSCLC is 15.8% [138]. Furthermore, *STK11/LKB1* and *KEAP1* mutations are associated with an inert tumor immune microenvironment, with a reduced density of infiltrating cytotoxic CD8^+^ T cells and a neutrophil-enriched TME [139]. *STK11/LKB1*- or *KEAP1*-mutated NSCLC patients had a higher bTMB than wild-type patients, and *KEAP1*-mutated patients had a higher PD-L1 expression (TC ≥ 50%/IC ≥ 50%: 25.00% vs. 14.54%), while *STK11/LKB1*-mutated patients had lower PD-L1 expression (TC ≥ 50%/IC ≥ 50%: 7.89% vs. 15.90%) [136].

However, regardless of TMB and PD-L1 status, the presence of *STK11/LKB1* and *KEAP1* mutations is a poor prognosis factor in lung cancer. A retrospective study of 174 *KRAS*-mutated NSCLC patients, including 31% with *STK11*/*LKB1* co-mutation, demonstrated significantly lower ORR and PFS following PD-1 inhibitor therapy in *STK11/LKB1*-co-mutated patients compared with *STK11/LKB1*-wild-type patients [140]. In an analysis of 99 *KEAP1*-mutated lung cancer patients using an online database, patients with KEAP1 mutations had a shorter OS after ICI treatment [138]. Furthermore, in a pooled analysis of the OAK and POPLAR studies, *STK11*/*LKB1*-mutated lung cancer patients had relatively worse OS than wild-type patients receiving atezolizumab [136]. However, the docetaxel-treated group also showed poor OS in *STK11*/*LKB1*-mutated patients [136]. Papillon-Cavanagh et al. demonstrated that *STK11*/*LKB1* and *KEAP1* mutations were associated with a poor prognosis in both ICI- and chemotherapy-treated populations [141]. These data suggest that *STK11*/*LKB1* and *KEAP1* mutations are prognostic markers and not predictive biomarkers for the efficacy of ICIs. However, these are only retrospective analyses, and a prospective evaluation is required.

### 4.7. Other Rare Oncogenic Driver Mutations

Lastly, *ROS1* rearrangement, *HER2* mutation, and *RET* rearrangement are rare driver gene mutations and thus are targets for molecular targeted therapy [142,143,144]. There are only a few small-scale reports detailing the efficacy of ICIs in NSCLC patients with these mutations, but all of them showed limited efficacy [145].

## 5. Discussion and Conclusions

Many developing biomarkers for ICI efficacy are based on tissue samples. However, bronchoscopic or CT-guided needle biopsy is the most common sampling method in advanced or recurrent lung cancer, and therefore, the obtained samples tend to be small, and the samples may not be representative because of intratumoral heterogeneity. Furthermore, in immunotherapy, the nature of the TME and tumor cells is essential, and it is often challenging to assess the tumor status in these small samples. Biopsies for histological examination are often invasive and challenging to perform in advanced NSCLC depending on the patient’s general condition. Conversely, blood biomarkers are non-invasive, reproducible, and reflect systemic conditions, and thus are suitable for the assessment of dynamic markers to determine tumor response and immunological changes in real-time before radiological or clinical progression. However, it does not reflect the intratumoral characteristics or microenvironment, and therefore tends to be less predictive for the efficacy of ICIs than tissue samples.

Currently, the only FDA-approved biomarkers for ICIs are PD-L1, TMB, and dMMR/MSI-H. However, some reports suggest that TMB is not predictive of ICI efficacy, and dMMR/MSI-H has limited utility due to its low frequency in lung cancer. TILs are expected to have a robust predictive effect among the biomarkers under development, but sufficient tissue collection for a correct evaluation is an obstacle. Although NLR does not have strong predictive power by itself, it is the easiest biomarker to evaluate in clinical settings and is expected to be fruitful when used in combination with other biomarkers. Because HLA-I genotypes are analyzed at different levels among racial groups, we expect to discover new candidate genotypes for biomarkers by expanding the scope of future analyses. However, because individual biomarkers are insufficient, it is desirable to make a comprehensive judgment based on a combination of various biomarkers in clinical settings, considering serological tests, histological tests, and the patient’s background (including ECOG performance status, body weight, comorbidities, and adherence), as well as the time course of the disease. For this, machine learning is another promising approach. Benzekry et al. developed machine learning models by collecting clinical and hematological data and predicted the disease control rate of ICIs at the individual level [146]. Alternatively, instead of a simple combination of biomarkers, recent developments in high-throughput analyses such as next-generation sequencing and mass spectrometry may make it easier to identify a panel of biomarkers.

Biomarkers currently in development are mainly focused on PD-1/PD-L1 inhibitors; however, CTLA-4 inhibitors are also currently in use. High CTLA-4 expression is associated with epithelial-to-mesenchymal transition, which plays a crucial role in immune resistance and is a potent driver for the activation of an immunosuppressive network within the TME, including lung cancer, and other biomarkers may be needed for CTLA-4 inhibitors [147]. In addition, a tumor acquires resistance to ICIs therapy because of the loss of neoantigens, antigen presentation ability deficit by structural changes in MHC-I/II, decreased INF-γ by mutations in the JAK1 and JAK2, T cell exhaustion, TME alternation recruiting suppressor cells (MDSCs), tumor-associated macrophages (TAMs) and regulator T cells (Tregs) [148]. Biomarkers focused on the mechanism of resistance may also aid in the treatment of PD-1/PD-L1 inhibitor-resistant lung cancer.

In conclusion, lung cancer drug therapy has demonstrated a significant breakthrough with the advent of ICIs. However, many lung cancer patients cannot benefit from ICIs, and thus biomarker development is an urgent issue in the delivery of ICIs to appropriate patients.

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
