# Peer review of "Predictive Markers for Immune Checkpoint Inhibitors in Non-Small Cell Lung Cancer"

_jcm, 2022, doi:10.3390/jcm11071855_

Round 1

Reviewer 1 Report

I read with interest the Review Article “Predictive Markers for Immune Checkpoint Inhibitors in Non-small Cell Lung Cancer” by Ryota Ushio et al. The authors review the results of the most common investigated predictive biomarkers for immunotherapy in NSCLC, such as PD-L1, tumor mutation burden (TMB), such as tissue and serum biological markers.

In its entirety, the article is well written, and interesting for the scientific community. On the other hand, given the large complexity of the discussed issue, the article focuses only on some aspects of the analysed biomarkers, leaving undiscussed some of the most important unanswered questions. Thus, some parts needs to be deeply remodelled to include the overall current knowledge status, as indicated by below comments.

Paragraph 2.1. This paragraph discusses the role of PD-L1 as predictive biomarker for mono- or combination therapy of immune checkpoint inhibitors. PD-L1 is the only approved biomarker to date to address non-oncogene addicted NSCLC patients to a treatment or another, and this needs to be highlighted. Moreover, the paragraph results too focused on technical issues of evaluating PD-L1 expression (immunohistochemistry antibodies and platforms). A crucial point of PD-L1 expression and evaluation is the effective predictive value of this biomarker, as well as all the biological and clinical related limitations (sampling, tumor heterogeneity, clinical benefit related to expression score, prognostic value) (PMID: 29057235, 32944371, 31555517). This has to be discussed, while the technical part should be shortened

Paragraph 2.2. Several aspects of TMB testing should be better discussed. Apart from quantity of mutations, quality of mutations should be also discussed (PMID 30755690). The authors discuss which cut-off could be adopted to define TMB, as well as the issue of tissue and circulating TMB: several emerging aspects related to these issues have to be deeply considered and discussed (i.e. technical issues, correlation between TMB and PD-L1, TMB heterogeneity and clonal architecture) (PMID 26940869, 30082870, 30816954, 28420421).

Along the same paragraph, the authors discuss the entire review, considering tissue and serum biomarkers, as well as driver mutations. Canonical markers and driver mutations should be separated and not discussed along the same paragraph.

The efficacy of ICIs in oncogene addicted tumors is limited, and the targetable drivers demonstrated to have limited efficacy in this subset of patients. These mutations should be more slightly discussed (in a separated paragraph, as indicated above).

2.7.1 The authors discuss the role of serum biomarkers to predict responsiveness to immunotherapy, but did not mention important emerging data to this aim through integrating liquid and tissue biopsy (PMID 33007267).

  1. Along the discussion, the authors should better focus on which biomarkers are more promising and powerful to predict patient response to immunotherapy. Moreover, the authors discuss that a panel of biomarkers could be the best strategy to predict patients immunotherapy (see above PMID 33007267), and should discuss which could be the most powerful multiparametric approach, also considering the new insights brought by machine learning and artificial intelligence to build parameters algorithms (PMID 34944830) and the potential of combining immunotherapeutic drugs to simultaneously block multiple immune checkpoints, acting on tumor microenvironment (PMID 31531020).

Minor concerns:

Line 134: the authors should define what “small mutations” stand for, whether SNVs or other.

Line 135: TMB is usually investigated interrogating coding sequences, please modify “genome”.

Line 142: when discussing the relation of smoking habits and tumor mutations, consider highlight response to immunotherapy in relation to cigarette-associated mutations (PMID 25765070).

Line 166: The authors cite the FDA approval for F1CDx assay, but also MSK-Impact assay reached FDA approval and should be mentioned (PMID 29506529).

Author Response

Paragraph 2.1. This paragraph discusses the role of PD-L1 as predictive biomarker for mono- or combination therapy of immune checkpoint inhibitors. PD-L1 is the only approved biomarker to date to address non-oncogene addicted NSCLC patients to a treatment or another, and this needs to be highlighted. Moreover, the paragraph results too focused on technical issues of evaluating PD-L1 expression (immunohistochemistry antibodies and platforms). A crucial point of PD-L1 expression and evaluation is the effective predictive value of this biomarker, as well as all the biological and clinical related limitations (sampling, tumor heterogeneity, clinical benefit related to expression score, prognostic value) (PMID: 29057235, 32944371, 31555517). This has to be discussed, while the technical part should be shortened

→Thank you for your comments. As per your request, to emphasize the point that the effect of immune checkpoint inhibitors varies with the intensity of PD-L1 expression, we added data from the KEYNOTE-010 study to lines 92–96, and that from the OAK study to lines 106-110. Furthermore, to highlight the discrepancy between biopsy and surgical specimens regarding PD-L1 expression due to tumor tissue heterogeneity and differences between primary and metastatic lesions, we added information from two relevant articles to lines 121–125. We have also shortened the technical subsection (lines 111-118).

Paragraph 2.2. Several aspects of TMB testing should be better discussed. Apart from quantity of mutations, quality of mutations should be also discussed (PMID 30755690). The authors discuss which cut-off could be adopted to define TMB, as well as the issue of tissue and circulating TMB: several emerging aspects related to these issues have to be deeply considered and discussed (i.e. technical issues, correlation between TMB and PD-L1, TMB heterogeneity and clonal architecture) (PMID 26940869, 30082870, 30816954, 28420421).

→Regarding one of the technical issues, we added a description about the impact of neoantigen intratumor heterogeneity to lines 173–181.

We revised the description of CheckMate 568 to clarify the reason why the cut-off value was set to 10 mut/Mb using F1CDx instead of whole-exome sequencing, and added the following to lines 144–149 to emphasize the independence of TMB as a biomarker: "enhanced response was independent of PD-L1 expression". We also added the point that TMB analyzed by F1CDx and WES showed good correlation (lines 160–162). Finally, we added TMB data from retrospective analyses of the POPLAR and OAK studies to lines 153–156.

Along the same paragraph, the authors discuss the entire review, considering tissue and serum biomarkers, as well as driver mutations. Canonical markers and driver mutations should be separated and not discussed along the same paragraph.

→As per your request, we revised the text and categorized the paragraphs as “2. Programmed Death-Ligand 1”, “3. Predictive Biomarkers Beyond PD-L1 Expression”, and “4. Tumor/Specific Genotype.”

The efficacy of ICIs in oncogene addicted tumors is limited, and the targetable drivers demonstrated to have limited efficacy in this subset of patients. These mutations should be more slightly discussed (in a separated paragraph, as indicated above).

→We added a new paragraph entitled “3.5.5. Blood/Tissue Composite Biomarker” and described the DIREct-Pre scoring system, which combines tumor PD-L1 expression with pre-treatment ctDNA and circulating immune cell profiling.

The authors discuss the role of serum biomarkers to predict responsiveness to immunotherapy, but did not mention important emerging data to this aim through integrating liquid and tissue biopsy (PMID 33007267).

→As per your request, we shortened the relevant paragraph, “4. Tumor/Specific Genotype.”

Along the discussion, the authors should better focus on which biomarkers are more promising and powerful to predict patient response to immunotherapy. Moreover, the authors discuss that a panel of biomarkers could be the best strategy to predict patients immunotherapy (see above PMID 33007267), and should discuss which could be the most powerful multiparametric approach, also considering the new insights brought by machine learning and artificial intelligence to build parameters algorithms (PMID 34944830) and the potential of combining immunotherapeutic drugs to simultaneously block multiple immune checkpoints, acting on tumor microenvironment (PMID 31531020).

→We added a discussion about machine learning to lines 468–470 of the revised manuscript, and the effects of simultaneous use of CTLA-4 inhibitor on the tumor microenvironment to lines 474–479.

Minor concerns:

Line 134: the authors should define what “small mutations” stand for, whether SNVs or other.

→As per your request, we corrected the text as follows on lines 162–165: “the number of mutations (somatic single variant (SNV) and multinucleotide variant (MNV) and small inthe number of mutations (somatic single variant (SNV) and multinucleotide variant (MNV) and small insertions and deletions (indels))”.

Line 135: TMB is usually investigated interrogating coding sequences, please modify “genome”.

→As per your request, we corrected the relevant text to “megabase (Mb) of interrogated coding sequence.” on line 135.

Line 142: when discussing the relation of smoking habits and tumor mutations, consider highlight response to immunotherapy in relation to cigarette-associated mutations (PMID 25765070).

→As per your request, we corrected the text on lines 142–144 of the revised manuscript: “In particular, lung cancer is primarily caused by chronic exposure to carcinogens in cigarette smoke, and the efficacy of ICIs correlates with a molecular signature characteristic of cigarette carcinogen–related mutagenesis, certain DNA repair mutations, and the burden of neoantigens”.

Line 166: The authors cite the FDA approval for F1CDx assay, but also MSK-Impact assay reached FDA approval and should be mentioned (PMID 29506529).

→As per your request, we added details about the FDA approval of the MSK-IMPACT assay to lines 160–162.

Reviewer 2 Report

The manuscript is well organized and the topic is clear and extensive.

The review is a good point of view in pulmonary pathology.

It could be nice to add a small paragraph to analyze the possible mechanism of resistance and drug resistance panel.

Author Response

It could be nice to add a small paragraph to analyze the possible mechanism of resistance and drug resistance panel.

→Thank you for your kind comments. I have added about the resistance of immune checkpoint inhibitors (ICIs) therapy in the discussion part on lines on lines 479-484.

The mechanisms of resistance in ICIs therapy are very complex and largely unexplored. In addition, treatment targeting the resistance mechanisms of ICIs is still in the developmental stage, while molecular targeted therapy for driver gene mutations resistant to TKIs has been developing, such as EGFR-TKIs (T790M mutation, C797S mutation, and MET amplification). Regrettably, the detailed mechanism of ICIs resistance and treatment targeting the resistance requires a significant amount of discussion and thus is beyond the scope of this article.

Reviewer 3 Report

The review article by Ryota et al. compiled data on predictive marker other than PD-L1 cell surface receptor which is currently used for immune checkpoint inhibitor(ICI) efficacy in non small cell lung cancer patients. The authors suggest that the combination of tissue-based, blood based biomarkers, the time course of disease and also patient’s background should be used for the benefit of ICI efficacy for lung cancer patients. They presented interesting information, but there are some questions and suggestions as described below.

  1. The abstracts and conclusions mention about the advantages and disadvantages of using tissue or blood-based biomarker, that's why the authors recommend a combination of at least these two indicators for ICI efficacy, however, the significance of these two points is not clearly demonstrated in the text. The authors should change the head topic of the text to consistent the abstract and conclusions.
  2. In the abstract, the authors indicate that the patient's background should be considered for ICI efficacy. The author should explain more that what's that background?
  3. The authors provide information of many biomarker So, the author should be able to suggest which predictive biomarkers are the most interesting or worth paying attention to? The conclusions and suggestion should be more specific.
  4. According to the suggestion by authors, predictive markers based on tissue, blood, patient history, and duration of disease should be used together in the selection of ICI. It means biopsy and liquid biopsy need to be collected. What do the authors think about increasing the difficulty for patients?
  5. The technical terms should be consistent, e.g. in the abstract, the authors say tissue-based and blood-based biomarker, In the discussion and conclusion, biopsy and liquid biopsy were used, while the serum biomarker was used in the main text.

Author Response

The abstracts and conclusions mention about the advantages and disadvantages of using tissue or blood-based biomarker, that's why the authors recommend a combination of at least these two indicators for ICI efficacy, however, the significance of these two points is not clearly demonstrated in the text. The authors should change the head topic of the text to consistent the abstract and conclusions.

→Thank you for your comments. As per your request, we revised the final sentence of the abstract as follows: “In addition to the individual biomarkers, the development of composite markers, novel technologies such as machine learning and high-throughput analysis may make it easier to analyze multiple comprehensively”(lines 19–22). We also added information about blood/tissue composite biomarkers and machine learning to lines 304–310 and 468–470, respectively.

In the abstract, the authors indicate that the patient's background should be considered for ICI efficacy. The author should explain more that what's that background?

→We revised the corresponding part in “5. Discussion and Conclusion” as follows: "… patient's background (including ECOG Performance Status, body weight, comorbidities, adherence), as well as the time course of the disease” (lines 466–467). However, we removed these points from the abstract because we considered them to be inappropriate here.

The authors provide information of many biomarker So, the author should be able to suggest which predictive biomarkers are the most interesting or worth paying attention to? The conclusions and suggestion should be more specific.

→As per your request, we described interesting biomarkers in the revised manuscript, as follows: “Currently, the only FDA-approved biomarkers for ICIs are PD-L1, TMB, and dMMR/MSI-H. However, some reports suggested that TMB was not predictive of the efficacy, and dMMR/MSI-H has limited utility due to its low frequency in lung cancer. Among the biomarkers under development, TILs are expected to have a robust predictive effect, but sufficient tissue collection for correct evaluation is an obstacle. Although NLR does not have strong predictive power by itself, it is the easiest biomarker to evaluate in clinical settings and is expected to be fruitful when used in combination with other biomarkers. Since HLA-I genotypes are analyzed at different levels among racial groups, we expect to discover the new candidate genotypes for biomarkers by expanding the future analysis scope.” (Lines 455–464).

According to the suggestion by authors, predictive markers based on tissue, blood, patient history, and duration of disease should be used together in the selection of ICI. It means biopsy and liquid biopsy need to be collected. What do the authors think about increasing the difficulty for patients?

→Liquid biopsy is a dramatically developing technique with great promise, but at present, the indications and benefits of liquid biopsy are limited in clinical settings and are generally not essential. The term "serological test" does not refer to the use of blood to collect tumor cells or cell-free DNA, but to general blood tests (blood counts, biochemistry, and tumor markers).

The technical terms should be consistent, e.g. in the abstract, the authors say tissue-based and blood-based biomarker, In the discussion and conclusion, biopsy and liquid biopsy were used, while the serum biomarker was used in the main text.

→We standardized all technical terms to “tissue biomarker” or “blood biomarker”.

Reviewer 4 Report

Thank you for the chance you gave me to read this interesting study entitled “Predictive Markers for Immune Checkpoint Inhibitors in Non-small Cell Lung Cancer” by Ushio et al. In this review, the authors give an overview of the current understanding of predictive markers for the efficacy of ICIs including tumor mutation burden, DNA mismatch repair deficiency, microsatellite instability, CD8+ tumor-infiltrating lymphocytes, human leukocyte antigen class I, tumor specific genotype, body mass index, and serum biomarkers such as peripheral T-cell phenotype, neutrophil-to-lymphocyte ratio, interferon-gamma, and interleukin-8. This is a very interesting topic based on the significance of ICIs and the study is well written.

Some of them are:

Very high similarity rate (33%) based on the Turnitin.

Since the authors have included a clinical parameter (BMI) in their study, clinical-based scores (e.g ALI, PIOS etc) should also be presented  

line 10: Please, remove “anti-“ since it doesn’t make sense.

lines 31-38: Please provide refs.

lines 138-139: Please, rephrase.

Lines 236-239: This sentence is not necessary since it is obscure how it is associated with the aims of the study.

line 273: Please, replace “embrolizumab” with pembrolizumab.

Lines 277-280: This sentence is not necessary since it is obscure how it is associated with the aims of the study.

Lines 315-317: This sentence is not necessary since it is obscure how it is associated with the aims of the study.

Lines 327-331 This sentence is not necessary since it is obscure how it is associated with the aims of the study.

lines 352-355: Please, rephrase. It doesn’t make sense.

lines 379-381: Since the authors have decided to focus on the NSCLC, this sentence is out of their scope.

line 451: Please, replace the “low systemic inflammation” with “low-grade systemic inflammation”.

Author Response

Very high similarity rate (33%) based on the Turnitin.

→We have corrected this similarity by referring to the attached PDF file.

Since the authors have included a clinical parameter (BMI) in their study, clinical-based scores (e.g ALI, PIOS etc) should also be presented 

→We removed the paragraph about BMI because the inclusion of the ALI and PIOS scoring systems would have unbalanced the article, and there were also many overlapping points.

line 10: Please, remove “anti-“ since it doesn’t make sense.

→As per your request, we removed “anti-” from this sentence.

lines 31-38: Please provide refs.

→The paragraph after lines 31–38 includes the relevant references for these statements. However, for the sake of clarity, we combined them into one paragraph.

lines 138-139: Please, rephrase.

→We rephrased lines 138–139 as follows: “The immune system recognizes neoantigens as non-self immunogenic targets and activate and targeting T cells.”

Lines 236-239: This sentence is not necessary since it is obscure how it is associated with the aims of the study.

→As per your request, we removed lines 236–239.

line 273: Please, replace “embrolizumab” with pembrolizumab.

→We corrected “embrolizumab” to pembrolizumab.

Lines 277-280: This sentence is not necessary since it is obscure how it is associated with the aims of the study.

→As per your request, we removed lines 277–280.

Lines 315-317: This sentence is not necessary since it is obscure how it is associated with the aims of the study.

→As per your request, we removed lines 315–317.

Lines 327-331 This sentence is not necessary since it is obscure how it is associated with the aims of the study.

→As per your request, we removed lines 327–331.

lines 352-355: Please, rephrase. It doesn’t make sense.

→We rephrased this sentence as follows on lines 424–426: “An analysis of 99 KEAP1-mutated lung cancer patients using an online database, patients with KEAP1 mutations had a shorter OS after ICI treatment”.

lines 379-381: Since the authors have decided to focus on the NSCLC, this sentence is out of their scope.

→We removed lines 379–381.

line 451: Please, replace the “low systemic inflammation” with “low-grade systemic inflammation”.

→As mentioned above, we removed the paragraph referring to BMI.

Round 2

Reviewer 1 Report

I think the manuscript has been sufficiently improved and is now ready to be published.

Reviewer 3 Report

The revised manuscript is easier to follow based on feedback from the reviewers and I am satisfied with the author's responses to the reviewer's comments. This is  an interesting review article of biomarkers for ICI selection in NSCLC.

Reviewer 4 Report

Questions asked by me have been answered by the authors.